# Effect of herbal extract granules combined with otilonium bromide on irritable bowel syndrome with diarrhoea: a study protocol for a randomised controlled trial

Joong Il Kim,[1] Pumsoo Kim,[1] Jin-Hyun Lee,[2] Yoo-Jin Kim,[1] Na-rae Yang,[1] Myong Ki Baeg,[1] Ja Sung Choi,[1] Hye-Jung Kim,[1] Jayoung Kim,[1] Yun-Young Sunwoo,[1] Jung-Han Lee,[2] Hyekyung Ha,[3] Tae-Yong Park[1]

JIK and PK contributed equally.

[1]Institute for Integrative Medicine, Catholic Kwandong University International St. Mary's Hospital, Incheon, Republic of Korea
[2]Department of Rehabilitation Medicine of Korean Medicine, College of Korean Medicine, Wonkwang University, Iksan, Jeonbuk, Republic of Korea
[3]K-herb Research Center, Korea Institute of Oriental Medicine, Daejeon, Republic of Korea

**Correspondence to**
Dr Tae-Yong Park;
parktae9822@gmail.com

## ABSTRACT

**Introduction** Irritable bowel syndrome (IBS), known as a functional and organic gastrointestinal disorder, is a collection of symptoms that occur together and generally include pain or discomfort in the abdomen and changes in bowel movement patterns. Due to the limitations of conventional treatments, alternative IBS treatments are used by many patients worldwide. Samryungbaekchulsan (SRS), a herbal formula, has long been used for alleviating diarrhoea-predominant IBS (D-IBS) in traditional Korean medicine. Otilonium bromide (OB) is an antimuscarinic compound used to relieve spasmodic pain in the gut, especially in IBS. Although herbal formulae and Western drugs are commonly coadministered for various diseases in Korea, few clinical studies have been conducted regarding the synergic effects of these treatments for any disease, including D-IBS.

**Methods and analysis** This trial is a randomised, double-blinded, placebo-controlled, double-dummy, four-arm, parallel study. After a 2-week preparation period, 80 patients with D-IBS will be randomly assigned to one of four treatment groups consisting of SRS (water extract granules, 5 g/pack, three times a day) with OB (tablet form, one capsule three times a day) or their placebos, with treatment lasting for 8 weeks. Post-treatment follow-up will be conducted 4 weeks after the end of treatment. The primary outcome is the finding obtained using the Subject's Global Assessment of Relief method. The secondary outcomes are the severity of symptoms related to D-IBS, determined using a 10-point scale, and the change in symptoms.

**Ethics and dissemination** This trial has full ethical approval of the Ethics Committee of Catholic Kwandong University International St. Mary's Hospital (IS15MISV0033) and the Korean Ministry of Food and Drug Safety (30769). The results of the study will be disseminated through a peer-reviewed journal and/or conference presentations.

**Trial protocol version** IS15MISV0033 version 4.0 (25 July 2016).

## Strengths and limitations of this study

► This randomised controlled trial is the first Korean clinical trial approved by the Korean Ministry of Food and Drug Safety to investigate the efficacy and safety of Samryungbaekchulsan (SRS; herbal formula) and otilonium bromide (OB; Western drug) in diarrhoea-predominant irritable bowel syndrome (D-IBS).

► To evaluate the therapeutic effect and safety of the coadministration of SRS and OB, this study is proposed as a double-blind, double-dummy, four-arm, parallel-group randomised controlled trial.

► The primary outcome is the improvement in the patient's symptoms according to the Subject's Global Assessment of Relief, and the secondary outcome is the severity of D-IBS symptoms.

► This study is conducted with a small sample size (four arms, 20 participants per group), so further clinical trials with powerful sample sizes will be required to more adequately determine the effectiveness of SRS and/or OB in D-IBS.

**Trial registration number** KCT0001621 (approval date: 10 August 2015).

## INTRODUCTION

Irritable bowel syndrome (IBS), a chronic gastrointestinal (GI) disease and functional bowel disorder, is characterised by abnormal bowel habits and abdominal pain without structural, morphological and histological abnormalities. It affects approximately 20% of the population worldwide, according to population-based studies.[1–6] In addition, recent evidence suggests that IBS is not only a functional disease but also an organic disease with a complex of symptoms, including infection, immune activation, serotonin dysregulation,

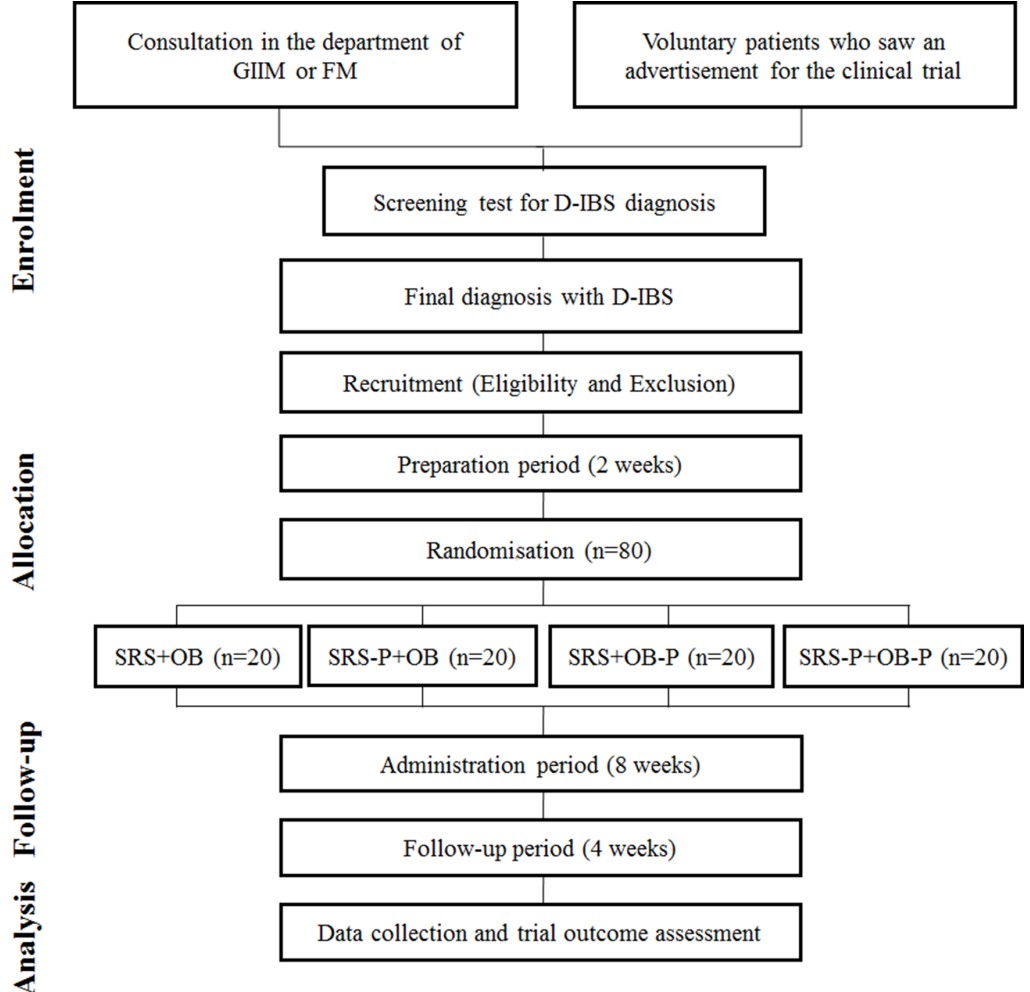

**Figure 1** Flow chart of trial. D-IBS, diarrhoea-irritable bowel syndrome; FM, family medicine; GIIM, gastrointestinal internal medicine; OB, otilonium bromide; P, placebo; SRS, Samryungbaekchulsan.

bacterial overgrowth, central dysregulation, brain–gut interaction and genetics.[4–8] Based on their predominant bowel habits, patients with IBS can be divided into diarrhoea-predominant IBS (D-IBS), constipation-predominant IBS or mixed IBS.[2 6 9] Recently, smooth muscle relaxants, bulking agents and antidiarrhoeal agents have been used in the treatment of IBS.[10] However, these methods are often not effective, and many patients with IBS also use complementary treatments, such as herbal medicine, acupuncture, psychological treatment and lifestyle correction.[2 11 12]

Samryungbaekchulsan (SRS; Shenlingbaizhu-san in traditional Chinese medicine; Jinryobyakujutsu-san in Kampo medicine), which was introduced in the famous classical compendium of herbal formulae 'Formulary of Peaceful Benevolent Dispensary', consists of 10 herbs (Atractylodes rhizoma Alba, Poria sclerotium, Dioscoreae rhizoma, Glycyrrhizae radix et rhizoma, Coicis semen, Nelumbinis semen, Platycodonis radix, Dolichoris semen, Amomi fructus and Ginseng radix). In traditional Korean and Chinese medicine, this herbal formula has been used to treat GI disease, and various studies have been published regarding the treatment of IBS with SRS.[13–18]

SRS has been shown to have antiacetylcholine and antibarium chloride effects and to suppress gastric secretion and small intestine motility.[13] However, no clinical trials have evaluated the efficacy of SRS for IBS when coadministered with a conventional drug.

Otilonium bromide (OB) is widely used for the prevention and reduction of symptoms in patients with IBS.[7 8] OB has been shown to inhibit L-type and T-type calcium channels, muscarinic receptors and tachykininergic responses in human cultured smooth muscle cells and rat colon strips.[19–21] According to a recent randomised controlled trial (RCT), OB (40 and 80 mg) is more effective for treating IBS than placebo, not only for the reduction of bloating and abdominal pain but also for protection from relapse owing to its long-lasting effect.[22 23]

According to the guidelines of the Korean Society of Gastroenterology, antispasmodic agents are recommended for the treatment of D-IBS (grade 1B),[24] and SRS is the preferred drug for diarrhoea treatment in Korean medicine.[18] Recently, several studies have been conducted to evaluate the therapeutic effect of the coadministration of herbal formulae and conventional drugs for IBS in China.[25 26] However, there has been no such clinical study

in Korea, even though both herbal medicine and conventional drugs are widely used for IBS treatment in the clinical field. Therefore, there is a need to evaluate the efficacy and safety of SRS, OB and the coadministration of both drugs.

In this study, we expect that the combined administration of OB and SRS will result in synergistic effects that improve D-IBS symptoms, including abdominal pain, discomfort (the main effect of OB) and diarrhoea-like stool patterns (the main effect of SRS).

## METHODS AND ANALYSIS
### Objectives
The purpose of this study is to evaluate the efficacy and safety of the coadministration of SRS and OB to patients with D-IBS.

### Necessity of the trial
Recently, herbal medicines and Western drugs have been coadministered for various medical therapies. This is particularly true in Korea, where approximately 76% of patients have used the combination of herbal medicines and Western drugs.[14] However, only a few studies have evaluated the interaction between Western and herbal medicines. Despite the need for combined treatment in patients with various diseases, there are no standard clinical guidelines or scientific evidence regarding coadministration because of a lack of research and data.

D-IBS is a disease that is difficult to cure, and many patients require treatment with both Western and herbal drugs to experience an improvement in their symptoms. Therefore, it is very important to conduct a clinical study regarding the efficacy and safety of the combined administration of SRS and OB in patients with D-IBS.

### Basis of drug selection
According to guidelines for IBS treatment, the OB recommendation level is high (ie, the level of evidence is moderate and many experts recommend using it), and it is the most reasonable treatment option.[25] SRS is the herbal formula used most often by clinical Korean and Chinese medical doctors for abdominal pain, abdominal discomfort and chronic diarrhoea, which are typical symptoms of D-IBS.[15–17 24] Based on these studies and clinical applications, OB and SRS were selected for this trial.

### Hypothesis
We hypothesise: (1) that coadministration of SRS and OB will have a more positive effect on D-IBS symptoms than either SRS or OB alone; (2) that after 8 weeks of treatment with SRS, OB or both drugs, the Subject's Global Assessment of Relief and severity of D-IBS symptoms will be more improved than after treatment with placebo and (3) that coadministration of SRS and OB will be safe for patients with D-IBS.

### Design
This study is a placebo-controlled, double-blind trial. Eighty patients will be enrolled and randomly assigned to four separate arms of the trial. The trial will be conducted at Catholic Kwandong University International St. Mary's Hospital, Incheon, South Korea.

The study subjects will be required to participate in a 2-week preparation period (weeks −2 to 0) and an 8-week drug-administration period (weeks 0–8). The patients will visit four times, on weeks 0, 4, 8 and 12. The primary outcome measurement will be evaluated at weeks 2, 4, 6, 8 and 12, and the secondary outcome measurement will be evaluated at weeks 0, 2, 4, 6, 8 and 12. Participants will be assessed for efficacy and safety of the treatment through visits (0, 4, 8 and 12 weeks) and telephone questionnaires (2 and 6 weeks). In addition, laboratory tests will be conducted at 0, 4 and 8 weeks. According to the intervention protocols, the clinical research coordinator (CRC) will encourage patients and participants to complete follow-up at every visit.

During the drug-administration period, SRS or its placebo will be given as one pack of water extract granules three times a day, 30 min before each meal. OB or its placebo will be given as one tablet three times a day, 30 min before each meal.

Participants will be divided into four groups: (1) SRS and OB group, (2) SRS and placebo-OB group, (3) placebo-SRS and OB group and (4) placebo-SRS and placebo-OB group. The entire flow chart of the study is shown in figure 1.

This study received the first approval given by the Korean Ministry of Food and Drug Safety (KMFDS) for a study of the coadministration of a conventional drug and a Korean traditional herbal medicine (approval number: 30769).

Potential participants will be provided with sufficient information about the study, and only those who agree to the study protocol and sign the consent form will be enrolled. The study will be implemented according to Good Clinical Practice and the revised version of the Declaration of Helsinki.

### Sample size calculation
This trial is a study for a new therapeutic regimen of SRS and OB combination treatment. In a similar study conducted previously by using coadministration of a probiotics mixture and an herbal medicine (placebo-controlled, double-blind and randomly four-arm trial), each arm consisted of at least 15 patients, and the total sample size was more than 60 participants.[27] In other similar studies, clinical trials have been conducted with a total of 60–64 participants, consisting of 12–30 patients per group.[28–30] Based on these previous studies, the minimal clinical significance of the trial can be obtained when only 16 participants complete the trial. Because the dropout rate is expected to be 20%, each trial group will be made up of at least 20 participants. Therefore, the total sample size must be more than 80 participants.

## Inclusion criteria

The following patients would be included: (1) patients aged 18–75 years; (2) patients satisfying the Rome III criteria of IBS (recurrent abdominal pain or discomfort that began at least 6 months before the clinical trial, with a rating of more than 3 on a 10-point scale, frequency of at least 3 days/month in the last 3 months and associated with two or more of the following: improvement associated with defecation, onset associated with a change in frequency of stools and onset associated with a change in form of stool)[1 9]; (3) patients satisfying the Rome III criteria of IBS, whereby D-IBS type is defined as loose/mushy or watery stools—Bristol Stool Form (BSF) Scale 6–7 ≥2 times/week[1 30]; (4) patients with a negative urine pregnancy test (in women of a childbearing age, within 7 days before the clinical trial) and who consent to using contraception during the administration period; (5) patients with the ability to read the symptom questionnaire and understand it and (6) patients who agree with the clinical plan and voluntarily sign the institutional review board (IRB)-approved documents.

## Exclusion criteria

The exclusion criteria are as follows: (1) chronic liver disease (cirrhosis, chronic hepatitis B or C); (2) chronic renal failure or renal impairment (serum creatinine ≥2 times the upper limit of normal); (3) liver dysfunction (aspartate aminotransferase (AST)/alanine aminotransferase (ALT) ratio ≥3 times the upper limit of normal); (4) diabetes (haemoglobin A1c >8% or not controlled by diet or medication), hypertension (≥160/100 mm Hg), thyroid dysfunction (exceptions in the following cases: if the disease is controlled by drug administration with a stable dose for 12 weeks before the screening, and the drug dose remains constant during the trial period), clinically significant haematological, cardiac, pulmonary or neurological disorders, or other severe systemic disorders; (5) abnormal findings on colonoscopy or colonography within the last 5 years; (6) history of surgery that affects GI motility (ie, gastrectomy, colonic resection, hysterectomy, except for appendectomy); (7) GI disease characterised by the following symptoms within 6 months before the clinical trial: inflammation of or ulcer in the oesophagus, stomach or duodenum; gastro-oesophageal reflux disease (at the discretion of the examiner, subjects may enrol if the disease is chronic and not acute); GI bleeding; GI stenosis or closure; infectious diarrhoea; inflammatory bowel disease (ie, Crohn's disease, diverticulitis, ulcerative colitis, infectious enteritis, ischaemic colitis); pancreatic insufficiency; biliary abscess; (8) mental illness or an addiction to drugs or alcohol; (9) severe systemic organ diseases such as cancer, autoimmune disease, stroke (although patients in remission for more than 5 years from a cancer unrelated to the GI tract can participate in the trial); (10) pregnancy, breast feeding and unwillingness to use contraception during the trial; (11) a history of taking the following drugs within 2 weeks before the trial: antibiotics, analgesics, antidepressants, antianxiety agents, anti-inflammatory agents, antiulcer agents, antigastric secretion inhibitors, laxatives, antispasmodics, antacid agents, GI stimulants, prostaglandin agents and corticosteroids; (12) use of medications that do not match the intent of this trial or have a clinical interaction with SRS or OB; (13) lactose intolerance (not controlled by food); (14) glaucoma; (15) participation in other clinical studies within 30 days prior to clinical screening and have received other clinical trial medications (including placebo); (16) hypersensitivity to clinical trial medicines and (17) other reasons considered inappropriate for participation in clinical trials.

## Recruitment, randomisation, blinding and non-blinding

Advertisements for the study will be posted at the Catholic Kwandong University International St. Mary's Hospital using banner advertisements and posters. We also plan to post the advertisements on the hospital home page, in the monthly hospital magazine, at the internet café and on the subway at least five times. All these advertisements will be done with the approval of the IRB.

After recruitment, patients will be randomised to one of the four trial arms. Randomisation will be performed using a list of block randomisations made by statisticians, independent of this clinical trial. A researcher who is not involved in the process of drug prescription and evaluation will execute the randomisation process.

During the treatment period, the patients and all researchers (investigators and CRCs) will be blinded, except the researchers who conducted the randomisation procedure. In addition, the researchers will not know the kind of medication that is being administered to the patient or any other information that could lead to bias.

Allocation concealment will be performed during the process of medication administration. The management pharmacist, who will not be involved in analysing the results of the trial, will use the randomisation table to label the medication with the assignment number and provide the medication. The patients will take the drugs contained in the opaque envelope, which will be labelled with the assignment number.

Non-blinding will be performed only in the following situations: (1) if the trial is terminated and there is a need for statistical analysis; (2) if a serious medical emergency occurs and information about the medication is needed and (3) if the chief investigator determines that non-blinding is necessary.

The entire process of the study will be directed by the authorised clinical research organisation (CRO), DreamCIS, Seoul, Korea.

## Intervention

The herbal formula SRS is commonly used by Korean traditional medical doctors for the treatment of diarrhoea and related diseases.[13–18] SRS used in this study (Samryungbaekchulsan granule; Han Kook Shin Yak, Nonsan, South Korea) is extracted with water and mixed with starch and lactose in accordance with Korean Good

**Table 1** Ingredients in the herbal formula Samryungbaekchulsan

| Scientific name | Part used | Grams per day |
| --- | --- | --- |
| Atractylodes rhizoma Alba | Root | 3.0 |
| Poria sclerotium | Dried core | 4.0 |
| Dioscoreae rhizoma | Root | 3.0 |
| Glycyrrhizae radix et rhizoma | Root | 1.5 |
| Coicis semen | Seed | 8.0 |
| Nelumbinis semen | Fruit | 4.0 |
| Platycodonis radix | Root | 2.5 |
| Dolichoris semen | Seed | 4.0 |
| Amomi fructus | Fruit | 2.0 |
| Ginseng radix | Root | 3.0 |

Manufacturing Practice. Production of SRS is regulated and allowed by the KMFDS. SRS contains 10 herbs, listed in table 1. Placebo-SRS is made from cornstarch and has the same taste, shape, colour and similar scent as SRS. SRS and placebo-SRS will be sealed in identical aluminium bags with the same labelling. These herbal drug packages will be distributed by an independent pharmacist in an isolated room. According to the recommended dosing methods of KMFDS, after SRS or placebo-SRS is dissolved in boiled water, the subjects will take them 30 min before each meal (5 g/pack, three times a day).

OB (Menoctyl Tab; Dong Hwa Pharm, Seoul, Korea) is a tablet containing OB 40 mg. Placebo-OB is a tablet made of Ludipress with the same taste, shape and colour as OB. Patients will be instructed to swallow one tablet of OB or placebo-OB 30 min before each meal over the administration period. The drugs used in the trial (SRS, OB, placebo-SRS, placebo-OB) have been approved by the KMFDS. Patients will be asked at the end of the study whether the drugs that they have been taking are either real or placebo, to measure the success of blinding. Compliance will be confirmed by counting returned SRS packets or OB tablets. Every patient will be instructed to write down any adverse events during the administration period, and these records will be evaluated at the follow-up. At the patient visits (at 0, 4 and 8 weeks), we will perform a check of vital signs, physical examination, ECG and laboratory tests to evaluate safety for the patients. All side effects will be reported to the investigator.

### Rescue therapy and concomitant medications

Participants are prohibited from taking any medication that could affect D-IBS during the entire course of the trial. However, if serious side effects occur or symptoms worsen, investigators will provide quick and appropriate treatment (including drug administration), based on their medical judgement. In this case, the direct investigators will record the treatment or drug administration and report to the IRB within 24 hours to determine whether the trial should be continued or discontinued.

### Outcome measurements

#### Primary outcome

The primary outcome will be measured via a questionnaire that confirms subjective symptom improvement. Participants will respond a total of five times during and after the administration period (at 2 and 6 weeks via telephone, and at 4, 8 and 12 weeks during direct visits) to the following question: Subject's Global Assessment of Relief—'How much do you think the symptoms of D-IBS have improved compared to before the clinical trial?' According to the answers given by the patients to this survey, the improvement is given a score of 0, 1, 2, 3 or 4.

#### Secondary outcomes

At 2, 4, 6, 8 and 12 weeks from the start of the treatment period, patients will be assessed for severity of D-IBS symptoms (abdominal pain, abdominal discomfort, satisfaction of defecation, frequency of abdominal pain, quality of life) using a Likert scale ranging from 0 to 10.[31] Patients will also be instructed to record the number of defecations per day, BSF and the degree of force used in bowel movements at 0, 2, 4, 6, 8 and 12 weeks.[32]

### Safety assessments

The investigator will record all adverse events that occur during the trial along with any concomitant medications in the case reports. When an adverse event occurs, the investigator will record the symptoms and signs of the adverse reaction, the duration (start and end date), severity, course, outcome, significance, causality by the trial drug and any action taken in relation to the adverse event. In the case of concomitant medicines, composition, dosage, duration of administration and reason for medication will be recorded in detail. Symptoms that existed before the start of the clinical trial will not be recorded as adverse events.

### Laboratory tests

The purpose of laboratory tests in this study is to evaluate the safety and pharmacokinetic profiles of SRS and OB. Because of the small number of patients assigned to this study and the different baseline values for each individual patient, we will not perform a detailed statistical analysis. However, if significant abnormal results are found in the laboratory tests, the clinical laboratory results will be described, and the connection to the drug will be considered.

Blood tests will be conducted at 0, 2, 4 and 8 weeks, and urine tests will be performed at 0 and 4 weeks during the clinical trial. In the case of a participant who drops out or the early termination of the clinical trial, blood and urine tests will be performed on the last day of the trial.

Using the test results (comprehensive verification of vital signs, laboratory tests, ECGs and physical examinations) for each individual patient, the investigator will determine adverse reactions caused by the drugs, and statistical tests will be conducted as needed on clinically significant parameters.

### Test of human derivatives

IBS is known to be closely related to stress.[12] Therefore, we plan to quantify factors associated with stress or that have been reported to worsen IBS symptoms as follows: (1) cortisol, (2) corticotropin-releasing hormone, (3) serotonin, (4) group I cytokines (10 species), (5) group II cytokines (14 species) and (6) growth factors (3 species).[33 34] Serum will be separated from blood specimens collected at 0, 4, 8 weeks, stored frozen and transferred to the Korea Institute of Oriental Medicine for examination.

### Early termination or dropout

The criteria for early termination or dropout are as follows: (1) administration of other drugs that are expected to affect the safety and efficacy of the clinical trial drugs; (2) request of participants to discontinue the clinical trial or withdrawal of the trial agreement; (3) occurrence of significant adverse drug reactions or events that preclude continuing the trial, according to the judgement of the clinical investigator; (4) new discovery of violations of significant clinical trial protocol during clinical trial; (5) less than 70% trial drug compliance and (6) any other reason that the clinical trial should be discontinued at the discretion of the clinical trial manager or investigator.

### Data collection, access and management

The data of this trial will be managed according to the standard work instructions of Catholic Kwandong University International, St. Mary's Hospital. Other contents not specified in the trial protocol shall follow the standards of the International Council for Harmonisation of Technical Requirements for Pharmaceuticals for Human Use (ICH) guideline for Good Clinical Practice and Korean Good Clinical Practice guideline.

The source document is recorded immediately when the data are collected. After the source document input is completed, it will be recorded in the case report form (CRF). All the documents will be kept safe, so that the data can be verified by the relevant government agencies and the IRB. Only research colleagues and others who have been approved by the principal investigator will have access to all data obtained from this trial.

### Statistical analysis

The statistical analysis will be conducted by blinded professional statisticians. To evaluate the efficacy of the study, both an intent-to-treat (ITT) analysis (primary analytical method) and a per-protocol (PP) analysis (secondary analytical method) will be performed. The ITT analysis will be conducted for all patients who were followed up at least once after randomisation and will be able to be evaluated for effectiveness. The PP analysis will be performed for patients who complete the 8-week treatment course and meet the inclusion and exclusion criteria.

To analyse the continuous variables in the demographic information and pretreatment characteristics between groups, a one-way analysis of variance (ANOVA) and Kruskal-Wallis test will be conducted. An ANOVA and Kruskal-Wallis test will also be used to test for symptom improvement compared with baseline (0 week) and the variables for analysis that include the following: (1) subjective symptom improvement (primary outcomes) and (2) severity of symptoms related to D-IBS (abdominal pain, abdominal discomfort, satisfaction of defecation, frequency of abdominal pain and quality of life). All values will be shown as mean±SEM. Comparisons of outcome assessments between the four groups will be performed using the paired t-test or Wilcoxon signed-rank test.

If the Subject's Global Assessment of Relief on the Likert scale is '0' or '1' for more than 2 weeks between weeks 8 and 12, then the symptoms will be considered to have improved sufficiently. Based on this, we will calculate the symptom relief rate by dividing the number of participants who exhibited sufficient symptom relief by the total number of participants in the trial and present descriptive statistics. The CATMOD procedure will be used to evaluate the improvement of symptoms compared with baseline. Comparisons between the four groups will be conducted using the $\chi^2$ test or Fisher's exact test. These statistical analyses will also be performed at 2, 4, 6, 8 and 10 weeks from baseline. The Cochran-Mantel-Haenszel test will be used to confirm the difference between the intervention groups at each time point.

A statistical analysis will also be conducted to evaluate safety. The incidence of adverse events and abnormality of experimental results will be analysed statistically according to the groups, and non-parametric methods will be applied as necessary using a paired t-test for continuous data, McNemar's test for categorical data, Fisher's exact test for adverse events and a generalised estimating equation for clinically significant changes between groups.

All statistical analyses will be conducted using SAS software, V.9.1.3 (SAS institute). A P value <0.05 will be considered statistically significant.

### Quality control and data monitoring

To ensure the results and quality of the clinical trial, assessments and monitoring will be performed by the CRO, DreamCIS, which is independent from the sponsor and competing interests. During the trial, the CRO will regularly monitor whether the study is proceeding in accordance with the protocol through the use of all related documentation, including trial master files, CRFs, informed consent forms and adverse event reports.

## DISCUSSION

Herbal medicine is a recognised effective treatment method based on clinical experience accumulated over thousands of years and more recently gained attention as an alternative therapy for treating various diseases that are difficult to treat or have unknown causes, such as IBS.[5 7 10 26 35] SRS is frequently used for abdominal

pain and diarrhoea in Korean and Chinese traditional medicine. According to experimental research, SRS has an anticholinergic effect that improves intestinal digestion and absorption, and inhibits intestinal movement, resulting in the prevention of diarrhoea and abdominal pain.[13] In addition, clinical studies have shown that the use of SRS in patients with D-IBS is effective for improving diarrhoea.[36 37] Some studies have reported that coadministration with SRS and trimebutine maleate or SRS and paroxetine is more effective than trimebutine maleate alone for the improvement of IBS symptoms.[15 16 25] Other studies have reported that SRS exerted more beneficial effects on patients with IBS compared with Smecta or probiotics.[17 37]

However, although there is a growing interest in the use of herbal medicines such as SRS in IBS, and related studies are continuously being published, there have been only a few well-designed RCTs.[22 23 26] To overcome the limitations of the previous studies, this trial was designed as a double-blind, double-dummy, four-arm, parallel-group RCT. The efficacy of SRS will be evaluated via the Subject's Global Assessment of Relief and the improvement in symptoms associated with bowel movements and abdominal pain. The results of these assessments are expected to provide distinct clinical evidence for the use of SRS in D-IBS treatment.

OB is a representative conventional medicine used for D-IBS treatment.[8 21 23 38 39] It is a spasmolytic agent that exerts its activity primarily in the distal GI tract through the inhibition of Ca++ flux and direct activation of contractile proteins in the smooth muscle.[20 21] OB has also been shown to reduce hypermotility and modulate visceral sensation, the effect of which is to improve D-IBS symptoms.[39 40] RCTs using OB for the treatment of D-IBS have been conducted. In these studies, treatment with OB resulted in a greater improvement in symptoms related to D-IBS (defecation frequency, regular intestinal habits, reductions in frequency of diarrhoea, abdominal pain and discomfort, severity of abdominal bloating) and better protection from symptom relapse compared with placebo.[22 23 38]

IBS is a disease with a high rate of progression and frequent recurrence. For this reason, patients with IBS often require a variety of treatment methods.[2 6 10] Among these methods, the combination of herbal and conventional drugs is widely used and is more effective for improving IBS symptoms than conventional drugs alone.[2 35] However, there are several limitations to the coadministration of herbal and conventional drugs in IBS. Specifically, there is a high risk of bias in related studies, and there are insufficient clinical guidelines regarding the interaction of herbal and conventional drugs in D-IBS, including SRS and OB. Therefore, it is necessary to perform well-designed RCTs to evaluate the coadministration of herbal and conventional drugs.

Because only a few studies have evaluated the combination of herbal and conventional drugs, our proposed study will significantly contribute to overcoming the limitations of previous studies and provide basic medical evidence for the coadministration of SRS and OB for D-IBS treatment. Therefore, this RCT has several characteristic features. First, this trial will be the first to investigate the efficacy of SRS and OB combination therapy for D-IBS. The results of previous studies suggested that the combination of OB and SRS will improve two major D-IBS symptoms at the same time: abdominal pain and discomfort (major effect of OB) and diarrhoea (main effect of SRS). This study will assess this hypothesis and provide a basis for the use of SRS and OB combination therapy for D-IBS. Second, the participants in this trial will be grouped into four arms, making it possible to compare the efficacy of combined treatment with the efficacy of each individual treatment. Third, this trial will evaluate the safety of the subjects using several assessments, including a periodic assessment of subjective symptoms, physical measurements and laboratory testing. The results of the safety analysis may provide evidence for the safety of treatment with the combination of herbal and conventional drugs in D-IBS. Fourth, this trial was the first clinical study related to combination treatment with Western and traditional herbal medicine in Korea to receive approval from the KMFDS. Through the systematic management of the trial under the auspices of the KMFDS, a high-quality trial will be performed, and the results of this study will provide information for use in further clinical applications.

Although our trial is designed with a small sample size, it has a creative and systematic research design, and the results will be useful basic clinical evidence in the field of integrated herbal and conventional drugs for the management of D-IBS.

## Trial status

The research plan and design began in April 2015.

The study received the IRB's final approval after two deliberations. The first submission date was 1 July 2015. The result was a request for revision and supplementation, which was received on 13 July 2015. We filed an application for re-review on 6 August 2015, supplemented with the relevant information, and received the IRB's final approval on 10 August 2015. We also requested that the KMFDS review the trial plan on 14 September 2015 and received approval for the clinical research on 21 December 2015.

The first participant was randomised on 29 March 2016. Recruitment for the study is ongoing. The primary completion date for this trial is anticipated to be 31 December 2017, and the study completion date is expected to be 28 February 2018.

## Ethics
### Protocol amendments

An investigator who wishes to change the protocol of the clinical trial plan should first discuss it with the trial director. Thereafter, the examiner should obtain prior approval from the IRB for changes to the protocol (except to prevent immediate injury to the patient, in which case

it will be reported to the IRB later). However, when a dangerous situation occurs and immediate treatment is needed, the investigator may report the protocol change to the IRB at a later time.

## Consent

Patients recruited through the announcement will receive a full explanation of the details of the trial in an easy-to-understand manner. The patients and investigators will have a mutual question and answer time before the patient is asked to sign the clinical trial consent. Through these processes, if the patient agrees to participate in the trial, a signed consent form will be obtained.

## Confidentiality

All patient information will be anonymised with initials or symbols, and all related investigators will keep the trial results confidential. The trial director will keep the signed consent forms, and prepare a list that will be used to confirm the patient identities.

## Post-trial care

If the patients experience unexpected accidents or injuries, appropriate compensation will be made by the Federal Insurance Company Korea (12th Floor Ferrum Tower 66 Suha-dong, Jung-gu, Seoul, 100210, South Korea), according to the patient compensation rules of the trial. Additionally, in the case of an emergency, the patients will receive appropriate medical care at the Catholic Kwandong University International, St. Mary's Hospital.

## Dissemination

The results of the trial will be disseminated through a peer-reviewed journal and/or conference presentations.

**Contributors** JIK, PK, J-HyL and J-HaL contributed to the protocol design and writing of the manuscript. HH is responsible for monitoring the process of the trial. N-rY, Y-JK, MKB, JSC, H-JK, J-YK and Y-YS provide advice on the process of the trial. T-YP is responsible for writing the manuscript and also managing and supervising the clinical research. All authors approved the final manuscript.

**Funding** This work was supported by a 'Construction of Scientific Evidences for Herbal Medicine Formulas (K15250)' grant from the Korea Institute of Oriental Medicine.

**Competing interests** None declared.

**Patient consent** Obtained.

**Ethics approval** This trial has been approved by the Institutional Review Board and Ethics Committee of Catholic Kwandong University International Saint Mary's Hospital (approval no IS15MISV0033; Clinical Research Information Service (www.cris.nih.go.kr) protocol identification number is KCT0001621 and the Korean Ministry of Food and Drug Safety (approval number 30769).

**Provenance and peer review** Not commissioned; externally peer reviewed.

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
