## [Reviewer comments · BMJ Open]

ARTICLE DETAILS

TITLE (PROVISIONAL)	Effect of herbal extract granules combined with otilonium bromide on irritable bowel syndrome with diarrhea: a study protocol for a randomized controlled trial
AUTHORS	Kim, Joong Il; Kim, Pumssoo; Lee, Jin-Hyun; Kim, Yoo-Jin; Yang, Na-rae; Baeg, Myong Ki; Choi, Ja Sung; Kim, Hye-Jung; Kim, Ja-Young; Sunwoo, Yun-Young; Lee, Jung-Han; Ha, Hyekyung; Park, Tae-Yong

VERSION 1 – REVIEW

REVIEWER	Dr Katja Boehm Faculty of Health Department of Psychology and Psychotherapy Chair of Research Methods and Statistic in Psychology University Witten/Herdecke Germany Germany
REVIEW RETURNED	05-Aug-2017

GENERAL COMMENTS	Review for: "Effect of herbal extract granules combined with otilonium bromide on irritable bowel syndrome with diarrhea: study protocol for a randomized controlled trial" Many thanks for giving me the opportunity to review this study protocol. I have a few points to consider for a revision of the protocol. 1. The English language needs to be revised carefully. I have found numerous mistakes, for example on page 22, line 24 "if the patients agree to participate in the trial, the consent form will be written." It should say 'signed' not 'written'. Or another example "A management pharmacist who is not affect the results of the trial and analysis, will refer to the randomization list and administer the medication." It should say "... who is not affected by..."2. Blinding: how will the blinding of the researchers carrying out the analysis be ensured? Double-blinding is mentioned but not explained further.3. Introduction: The following sentence needs correcting "However, these methods are often not effective, and many IBS patients also use alternative treatments such as herbal medicine, acupuncture, psychological treatment, and lifestyle correction [2, 5, 6]." Usually, these therapies are applied as complementary, not alternative therapies.4. Intervention: "has the same taste, shape, and color as SRS." – what about scent?5. Could diet / liquid intake affect the effects of the herb? and/or medication?
---

REVIEWER	franco scaldaferrì Institute of Medical Pathology Catholic University of the Sacred Heart Fondazione Policlinico "A. Gemelli" Hospital Igo Gemelli, 8 - 00168 Roma
REVIEW RETURNED	20-Aug-2017

GENERAL COMMENTS	Please re-do the sample size calculation as a superiority trial (both treatment are better than 1, with the control being the OB alone). Please report more info about the herbal compound: clinical or preclinical studies, if available for the formulation as a whole or also as single compound. Please discuss how you selected the study drug dosage.
--

REVIEWER	Maria Giuliana Vannucchi University of Florence, Italy
REVIEW RETURNED	21-Aug-2017

GENERAL COMMENTS	The present clinical trial project, aimed to evaluate the efficacy of a combined therapy between traditional oriental medicament and a drug commonly used in the western countries, is theoretically interesting and culturally significant. From the theoretically point of view, this approach underlies the will to keep the patient interest at the first place independently on the proper traditions and praxis. From the cultural point of view, it represents an attempt to favor the mutual recognition between two different cultures. Regarding the content and the methodology present in the manuscript, I consider the protocols scientifically credible and presented in an appropriate context, the design is ethically and procedurally sound. However, I have two main concerns that the authors should consider in reviewing the text and the study protocol:  1. The authors should do an appropriate revision of the references quoted in the text. Indeed, the publications listed are mainly limited to authors and drugs of the eastern countries. The authors should extend the reference list to the numerous papers concerning OB. Furthermore, the way to quote the references is sometimes incorreced since papers on the same issue (n° 13 and 31) are quoted one in the introduction (13), the other in the discussion (31). There are some mistakes in quoting in the text the references n° 10,11 and 12 and in writing the references (i.e. n°26). The authors should control and correct. 2. The second concern regard the exclusion criteria. I suggest excluding any person that has ever used the herbal formula SRS, this is because of the way it is administered. In other words, I believe that any person that has tasted before the infusion can recognize whether the formula that is assuming during the trial is the drug or the placebo and it could affect the results. If the authors already considered this criterion, it should be added at the list.
--

	Minor points ABSTRACT Substitute 'simultaneously' with together. The last sentence of the Introduction is ambiguous, it seems to suggest that there are other studies on the present subject. INTRODUCTION Pag.7 The first row contains a repetition describing IBS. I would suggest the first definition also because an accurate revision of the data available on IBS has bring several authors to reconsider the possibility that local damages are present in IBS (therefore ii could not be a simple functional disease). Attention to the references 10, 11 ,12 Pag.8 Add reference n° 31 to 13. The last paragraph contains the same ambiguity of the last sentence in the Abstract. METHODS AND ANALYSIS Exclusion criteria See the main concern. 'Anti-inflammatory drugs': it is written twice Recruitment, Randomization, etc.. Last paragraph: 'who is not affect..'?'''''' DISCUSSION Pag.22. First paragraph: - First sentence: Remove the second 'disease'. - Second sentence: Integrate with the adequate references and remove the reference 31 that is not correctly quoted.
--	--

VERSION 1 – AUTHOR RESPONSE

Reviewer: 1

Point#1.

The English language needs to be revised carefully. I have found numerous mistakes, for example on page 22, line 24 “if the patients agree to participate in the trial, the consent form will be written.” It should say ‘signed’ not ‘written’. Or another example “A management pharmacist who is not affect the results of the trial and analysis, will refer to the randomization list and administer the medication.” It should say “... who is not affected by...”

[Answer]

As suggested, we have corrected the said sentences in the article (main draft, p. 12 and p. 22):

(1) Ethics section: Consent (main draft, p. 22)

Through these processes, if the patient agrees to participate in the trial, a signed consent form will be obtained.

(2) Recruitment, randomization, blinding, and unblinding section (main draft, p. 12)

The management pharmacist who will not be involved in analyzing the results of the trial, will use the randomization table to label the medication with the assignment number and provide the medication.

Point#2.

Blinding: how will the blinding of the researchers carrying out the analysis be ensured? Double-blinding is mentioned but not explained further.

[Answer]

As your comment suggested, we have added more information about double-blinding to the “Recruitment, randomization, blinding, and unblinding” section (main draft, p. 12):

During the treatment period, the patients and all researchers (investigators and clinical research coordinators) will be blinded, except the researchers who conducted the randomization procedure. In addition, the researcher will not know the kind of medication that is being administered to the patient or any other information that could lead to bias.

Point#3.

Introduction: The following sentence needs correcting “However, these methods are often not effective, and many IBS patients also use alternative treatments such as herbal medicine, acupuncture, psychological treatment, and lifestyle correction [2, 5, 6].” Usually, these therapies are applied as complementary, not alternative therapies.

[Answer]

As suggested above, we corrected “alternative” to “complementary”, as shown in the following sentence (main draft, p. 5):

However, these methods are often not effective, and many IBS patients also use complementary treatments such as herbal medicine, acupuncture, psychological treatment, and lifestyle correction.

Point#4.

Intervention: “has the same taste, shape, and color as SRS.” – what about scent?

[Answer]

Placebo-SRS and SRS are both prepared with cornstarch as the primary ingredient; the scents of the two drugs are very similar, but not identical. Therefore, we have added “similar scent” to that description. In addition, the placebo-SRS used in this study has also been approved for use in studies by the Korean Ministry of Food and Drug Safety (KMFDS). Therefore, we think that this slight difference in scent would not allow the patient to identify the type of drug.

As you suggested, we have edited the following sentences describing the intervention (main draft, p. 13):

Placebo-SRS is made from cornstarch and has the same taste, shape, color, and similar scent as SRS.

According to the recommended dosing methods of KMFDS, after SRS or placebo-SRS is dissolved in boiled water, the subjects will take them 30 min before each meal (5 g/pack, 3 times a day).

Point#5.

Could diet / liquid intake affect the effects of the herb? and/or medication?

[Answer]

To the best of our knowledge, no previous studies have suggested that diet/liquid affects the therapeutic effect associated with herbal medicines; traditionally, patients in Korea do not specifically limit their food intake when they take herbal medicines. Therefore, we believe that if there is an influence of diet/liquid on the treatment effect of herbal medicine, it is only mild.

Reviewer: 2

Point#1.

Please re-do the sample size calculation as a superiority trial (both treatment are better than 1, with the control being the OB alone).

[Answer]

This clinical trial is a pilot study protocol to explore the efficacy and safety of a complementary treatment combined with SRS and OB on D-IBS. We did not calculate the sample size as it was a superiority trial. However, similar to other pilot studies, we assumed a minimum number of patients per group of 20, considering a dropout rate of 20%. Therefore, we considered that the sample size was sufficient for a pilot study (main draft, p. 9).

Point#2.

Please report more info about the herbal compound: clinical or preclinical studies, if available for the formulation as a whole or also as single compound.

[Answer]

In the discussion section, we added sentences to describe the relevant clinical studies (main draft, p. 19).

In addition, clinical studies have shown that the use of SRS in patients with D-IBS is effective for improving diarrhea^{32 33}. Some studies have reported that co-administration with SRS and trimebutine maleate or SRS and paroxetine is more effective than trimebutine maleate alone for the improvement in IBS symptoms^{11 12 21}. Other studies have reported that SRS exerted more beneficial effects on IBS patients compared with Smecta or probiotics^{13 33}.

Point#3.

Please discuss how you selected the study drug dosage.

[Answer]

The dosage of the herbal drug used in this study (SRS; Samryungbaekchulsan granule®, Hankook shin yak, Co., Ltd, Nonsan, Korea) was based on the criteria of the Korean Ministry of Food and Drug Safety (KMFDS); the following relevant information has been added to the 'Intervention' section (main draft, p. 13):

According to the recommended dosing methods of KMFDS, after SRS or placebo-SRS is dissolved in boiled water, the subjects will take them 30 min before each meal (5 g/pack, 3 times a day).

Reviewer: 3

Point#1.

The authors should do an appropriate revision of the references quoted in the text. Indeed, the publications listed are mainly limited to authors and drugs of the eastern countries. The authors should extend the reference list to the numerous papers concerning OB.

[Answer]

As you mentioned, we have corrected all references based on the context of each sections and rearranged them in the quoted order. Further, we have integrated and added several references related to this manuscript.

Point#2.

Furthermore, the way to quote the references is sometimes incorreced since papers on the same issue (n° 13 and 31) are quoted one in the introduction (13), the other in the discussion (31).

[Answer]

As noted, we integrated references from the same issue #13 (Chmielewska-Wilkon D, et al., 2014) and #31 (Clave P, et al., 2011). The numbering of all references in the manuscript was rearranged to match the citation order.

Point#3.

There are some mistakes in quoting in the text the references n° 10,11 and 12 and in writing the references (i.e. n°26). The authors should control and correct.

[Answer]

As suggested, we removed the reference article #26 (Li YL, et al., 1991), which was not correctly quoted.

Point#4.

The second concern regard the exclusion criteria. I suggest excluding any person that has ever used the herbal formula SRS, this is because of the way it is administered. In other words, I believe that any person that has tasted before the infusion can recognize whether the formula that is assuming during the trial is the drug or the placebo and it could affect the results. If the authors already considered this criterion, it should be added at the list.

[Answer]

We believe this is reasonable and valid, but it is impossible to obtain a detailed drug history of patients with respect to the herbal formula SRS. In Korea, herbal medicines are consumed by many patients, but the specific details of herbal medicines prescribed by a Doctor of Korean Medicine are not provided to the patient or other doctors under Korean medical law and customs. However, as the placebo-SRS and SRS used in the study were manufactured to be similar in shape and taste, it was impossible for the patients to distinguish the different tastes.

Minor Point#1.

ABSTRACT

Substitute 'simultaneously' with together.

The last sentence of the Introduction is ambiguous, it seems to suggest that there are other studies on the present subject.

[Answer]

As suggested, we edited the following sentence in the introduction section of the abstract (main draft, p. 3):

Although herbal formulas and western drugs are commonly co-administered for various diseases in Korea, only few studies have been conducted regarding the synergic effects of these treatments for any disease, including D-IBS.

Minor Point#2.

INTRODUCTION

Pag.7

The first row contains a repetition describing IBS. I would suggest the first definition also because an accurate revision of the data available on IBS has bring several authors to reconsider the possibility that local damages are present in IBS (therefore ii could not be a simple functional disease).

Attention to the references 10, 11 ,12

[Answer]

As you mentioned above, we have added additional literature references in the 'abstract' and 'introduction' section to show that IBS is not simply a functional disorder, but a structural disease (main draft, p. 3 and p. 5):

(1) Abstract section (main draft, p.3)

Irritable bowel syndrome (IBS), known as a functional and organic gastrointestinal (GI) disorder, is a collection of symptoms that occur together and generally include pain or discomfort in the abdomen and changes in bowel movement patterns.

(2) Introduction section (main draft, p. 5)

Irritable bowel syndrome (IBS), a chronic gastrointestinal (GI) disease and functional bowel disorder, is characterized by abnormal bowel habits and abdominal pain without structural, morphological, and histological abnormalities. It affects approximately 20% of the population worldwide, according to population-based studies1-6. In addition, recent evidence suggests that IBS is not only a functional disease but also an organic disease with a complex of symptoms, including infection, immune activation, serotonin dysregulation, bacterial overgrowth, central dysregulation, brain-gut interaction, and genetics4-6.

Minor Point#3.

INTRODUCTION

Pag.8

Add reference n° 31 to 13.

The last paragraph contains the same ambiguity of the last sentence in the Abstract.

[Answer]

As noted, we corrected and added more references with similar issues considering all sections. We edited the final paragraph in the introduction section, as follows (main draft, p. 6):

According to the guidelines of the Korean Society of Gastroenterology, antispasmodic agents are recommended for the treatment of D-IBS (Grade 1B),¹⁹ and SRS is the preferred drug for diarrhea treatment in Korean medicine²⁰. Recently, several studies have been conducted to evaluate the therapeutic effect of the co-administration of herbal formulas and conventional drugs for IBS in China^{21 22}. However, there has been no such clinical study in Korea, even though both herbal medicine and conventional drugs are widely used for IBS treatment in the clinical field. Therefore, there is a need to evaluate the efficacy and safety of SRS, OB, and the co-administration of both drugs.

In this study, we expect that the combined administration of OB and SRS will result in synergistic effects that improve D-IBS symptoms, including abdominal pain, discomfort (the main effect of OB), and diarrhea-like stool patterns (the main effect of SRS).

Minor Point#4.

METHODS AND ANALYSIS

Exclusion criteria

See the main concern.

'Anti-inflammatory drugs': it is written twice

Recruitment, Randomization, etc..

Last paragraph: 'who is not affect..'?''''''

[Answer]

As indicated, we removed the repeated "Anti-inflammatory drugs" in the Exclusion section (main draft, p. 11). We edited the sentence with the phrase "... who is not affect..." in the "Recruitment, randomization, blinding, and unblinding" section as follows (main draft, p. 13):

The management pharmacist who will not be involved in analyzing the results of the trial, will use the randomization table to label the medication with the assignment number and provide the medication.

Minor Point#5.

DISCUSSION

Pag.22. First paragraph:

- First sentence: Remove the second 'disease'.

- Second sentence: Integrate with the adequate references and remove the reference 31 that is not correctly quoted.

[Answer]

As indicated, we removed the second repeat of "disease" in the sentence (main draft, p. 20). In addition, we corrected the references and included different references related to similar topics in all sections (main draft, p. 20), as previously mentioned.

VERSION 2 – REVIEW

REVIEWER	Katja Boehm Institute for Integrative Medicine Research and Teaching Center Herdecke University Witten/Herdecke Germany Gerhard Kienle Chair of Medicine, Integrative and Anthroposophic Medicine and Faculty of Health Department of Psychology and Psychotherapy Chair of Research Methods and Statistic in Psychology University Witten/Herdecke Germany Germany
REVIEW RETURNED	29-Sep-2017

GENERAL COMMENTS	The paper has been substantially improved and the protocol is ready for publication in my opinion.
--

REVIEWER	franco scaldaferrì università cattolica del sacro cuore, rome, italy
REVIEW RETURNED	01-Oct-2017

GENERAL COMMENTS	the paper improved significantly. please check the final destination of the new session: STRENGTHS AND LIMITATIONS OF THIS STUDY. furthermore, this sectio, in discussions, need more speculazio
--

REVIEWER	Maria Giuliana Vannucchi Department of Experimental and Clinical Medicine. University of Florence, Florence, Italy
REVIEW RETURNED	28-Sep-2017

GENERAL COMMENTS	Comments to the revised version I appreciate the efforts the authors made to satisfy the referees 'request. However there are two points the authors should take into consideration before the paper could be accepted for publication. 1) The authors answered to some of the request however, they missed one of the main request that is an adequate update of the references. Indeed, in spite what they write in the answer to the referee's comments, I did not see any new reference in the list. The authors have to fix this point. I would suggest some references: Cipriani et al. 2011 Neurogastroenterol Motil, Cipriani et al. 2015 Neurogastroenterol Motil, Traini et al. 2013, Neurogastroenterol Motil, Traini et al. 2014, PlosOne Traini et al. 2017, J Cell Mol Med Barbara et al. 2004, Gastroenterology Boeckxstaens et al. 2013, Int J Colonrectal Dis
---

	2) Reviewing the literature of the last 25 years I never found the term octylonium in any scientific paper. I strongly suggest the authors to go back to the previous term otilonium that is correct and shared by the entire scientific community.
--	---

VERSION 2 – AUTHOR RESPONSE

Reviewer: 1

Point#1.

- The paper has been substantially improved and the protocol is ready for publication in my opinion.

[Answer]

We appreciate your opinion.

Reviewer: 2

Point#1.

the paper improved significantly.

please check the final destination of the new session: STRENGTHS AND LIMITATIONS OF THIS STUDY.

furthermore, this section, in discussions, need more speculation

[Answer]

According to the BMJ Open guidelines and the editorial requirement: Point#1, the “STRENGTH AND LIMITATIONS OF THIS STUDY” section should include the content of the methods. In this sense, considering your suggestion, we have added the following sentences in the “STRENGTHS AND LIMITATIONS OF THIS STUDY” and the “DISCUSSION” section.

(1) STRENGTHS AND LIMITATIONS OF THIS STUDY section

To evaluate the therapeutic effect and safety of the co-administration of SRS and OB, this study is proposed as a double-blind, double-dummy, four-arm, parallel-group randomized controlled trial.

(2) DISCUSSION section

Because only a few studies have evaluated the combination of herbal and conventional drugs, our proposed study will significantly contribute to overcoming the limitations of previous studies and provide basic medical evidence for the co-administration of SRS and OB for D-IBS treatment.

In addition, we believe that the final destination of this study protocol is sufficiently described in the discussion section, including the statements in the final paragraph of the discussion section.

Reviewer: 3

Comments to the revised version

I appreciate the efforts the authors made to satisfy the referees' request. However there are two points the authors should take into consideration before the paper could be accepted for publication.

Point#1.

- The authors answered to some of the request however, they missed one of the main request that is an adequate update of the references. Indeed, in spite what they write in the answer to the referee's comments, I did not see any new reference in the list.

The authors have to fix this point.

I would suggest some references:

Cipriani et al. 2011 Neurogastroenterol Motil,
Cipriani et al. 2015 Neurogastroenterol Motil,
Traini et al. 2013, Neurogastroenterol Motil,
Traini et al. 2014, PlosOne
Traini et al. 2017, J Cell Mol Med
Barbara et al. 2004, Gastroenterology
Boeckxstaens et al. 2013, Int J Colorectal Dis
[Answer]

We apologize because we did not provide the proper answer. The following references were already added to the INTRODUCTION and DISCUSSION sections in the previous major revision:

1. Drossman DA, et al., Am J Gastroenterol 2000
2. Maxion-Bergemann S, et al., Pharmacoeconomics 2006
3. Shi J, et al., World J Gastroenterol 2008
4. Ko SJ, et al., Evid Based Complement Alternat Med 2013
5. Saha L., World J Gastroenterol 2014
6. Soares RL., World J Gastroenterol 2014
7. Chey WD, et al., JAMA 2015
8. Shin B, Lee B, Herbal Formula Science 2015
9. Whitehead AL, et al., Stat Methods Med Res 2016

From your suggestion, we have added the following references in the INTRODUCTION and DISCUSSION sections of this revision.

1. Boeckxstaens G, et al., Int J Colorectal Dis 2013
2. Traini C, et al., Neurogastroenterol Motil 2013
3. Traini C, et al., J Cell Mol Med 2017

Further, we have rechecked and rearranged the references in the quoted order. Finally, we have integrated and added 12 new references related to this manuscript.

Point#2.

- Reviewing the literature of the last 25 years I never found the term octylonium in any scientific paper. I strongly suggest the authors to go back to the previous term otilonium that is correct and shared by the entire scientific community.

[Answer]

According to your comment, we checked the terminology between 'octylonium' and 'otilonium'. Octylonium and otilonium are two names for the same drug, but otilonium has been more widely used in academia since the 2000s. Therefore, we decided to change the terminology from 'octylonium' to 'otilonium' and obtained approval from the IRB and KMFDS. We deeply appreciate your thoughtful comment.